



# Long-term aerosol mass concentrations in southern Finland: instrument validation, seasonal variation and trends

Helmi-Marja Keskinen[1,2], Ilona Ylivinkka[1], Liine Heikkinen[1], Pasi P. Aalto[1], Tuomo Nieminen[1,3], Katrianne Lehtipalo[1], Juho Aalto[1], Janne Levula[1], Jutta Kesti[1,4], Lauri R. Ahonen[1], Ekaterina Ezhova[1], Markku Kulmala[1], and Tuukka Petäjä[1]

[1]INAR/Physics, University of Helsinki, Finland

[2]Aerosol Physics Laboratory, Physics Unit, Tampere University, Tampere, FI-33720, Finland

[3]INAR/Forest Sciences, University of Helsinki, Finland

[4]Finnish Meteorological Institute, Helsinki, Finland.

10 *Correspondence to*: Helmi-Marja Keskinen (helmi-marja.keskinen@tuni.fi)

**Abstract.** Long-term high-quality aerosol particulate matter (PM) concentration measurements have been conducted in southern Finland at Station for Measuring Ecosystem-Atmosphere Relations (SMEAR II, Hyytiälä) with different, yet comparable measurement equipment since 1995. In this paper, the mass concentrations measured between 2005 and 2017 using three different independent methods: i.e. 1. the cascade impactor, 2. Differential Mobility Particle Sizer (DMPS) and Aerosol Particle Sizer (APS) and 3. Synchronized Hybrid Ambient Real-time Particulate Monitor (SHARP) are compared and analysed. First, the mass concentrations of the different size classes, i.e. PM1 (PM within sub-micrometer particle diameter size range), PM2.5 (sub-2.5 µm) and PM10 (sub-10 µm), are reported. These data were further cross-compared through a bivariate fitting method. The comparison revealed an excellent equivalence among the three methods with slopes approaching unity and reasonable intercepts ($\leq 1$ µg m$^{-3}$). An analysis of the seasonal variability of PM concentrations revealed that the mass concentrations were generally highest in summer in different size classes. The mean mass concentrations were 5.3, 5.4, and 6.5 µg m$^{-3}$ for PM1, PM2.5 and PM10, respectively. The 2$^{nd}$ highest loadings were attained in spring, which were ca. 80–88 % of those in summer. The lowest loadings were measured in autumn and winter, when the mass concentrations were ca. 74–78% of those in summer. Temperature had strong influence on the measured concentrations. While the high late spring and summertime temperatures promote secondary organic aerosol (SOA) formation and pollen emissions, the lowest wintertime temperatures enhance the need of residential heating processes yielding anthropogenic aerosol emissions (e.g. from traffic/industry/wood burning). The wintertime concentrations can also be expected to be influenced by boundary layer dynamics, which keep the PM emissions concentrated near Earth surface especially in winter. It is noteworthy that the mass concentrations were lower than those reported prior to 2005 (at SMEAR II). The descending trend (~-0.1-0.2 µg m$^3$y$^{-1}$) was clearly visible here for all PM size classes in spring, summer and winter, while the trend in autumn remained statistically insignificant. This might have resulted at least partly from more stringent EU air quality legislation.



## 1. INTRODUCTION


The air that we breathe consists of gases and particulate matter (PM). If the inhaled particles end up in our respiratory system, they may cause serious health hazards such as asthma or cardiovascular diseases (Pope et al., 2003). Tragic events, such as the Great Smog of London in December 1952 that led to the death of approximately 12 000 people, has evoked a need to monitor and regulate PM concentrations (Bell and Davis, 2001). Nowadays, PM concentrations and other

atmospheric variables are and will be reported online worldwide (https://www.who.int/gho/phe/outdoor_air_pollution visited 10/2020, (Laj et al., 2020)). With the increasing knowledge regarding the relationship between air pollution and mortality (Zeger et al., 2001), human activities have been regulated to achieve better air quality. Besides the adverse health effects, aerosol particles can also scatter or absorb radiation and participate in cloud formation and processing thus affecting the Earth's climate (IPCC, 2013). Hence, PM concentrations and their regulations are under the great interest.

The mass concentration of particles with (aerodynamic) diameter less than 1 μm is called PM1, less than 2.5 μm PM2.5, and less than 10 μm PM10. Primary PM consists mostly of particles from traffic (e.g. black carbon) and industry pollution or have natural source such as volcanic ash, desert dust and pollen. Secondary PM (mostly in within the accumulation mode, e.g. PM1) in the atmosphere is formed through new particle formation (Kulmala et al., 2013) or condensation or uptake of oxidized vapors in the atmosphere. These vapors include for example a myriad of organic species and sulfuric acid. The

oxidized, condensable products form in atmospheric oxidation from their volatile precursors. Over industrialized areas, the emitted sulfur dioxide ($SO_2$), forms sulfuric acid upon oxidation. Several volatile organic compounds (VOCs) are subjected to oxidation after emissions. These VOCs include both biogenic and anthropogenic vapors. Monoterpenes represent a group of VOCs emitted relatively efficiently in the boreal forest (Rinne et al., 2005). Their emission rates are boosted under warm temperatures (Guenther et al., 1993). Due to the atmospheric lifetime of aerosol particles spanning up to a week (Wagstrom

and Pandis, 2009), the PM measured even in pristine locations can contain significant amounts of long-range transported PM. Therefore, for example ammonium sulfate is a common constituent even within the boreal forest (Heikkinen et al., 2020b).

The size of atmospheric aerosol particles is perhaps their most critical parameter, both in terms of their climate (e.g. Dusek et

al., 2006) and health effects (Pope et al., 2003). In principle, the smaller the particles are, the deeper they will penetrate in human respiratory system and can thus end up also the other organs beside the lungs (Pope et al., 2003). The mass concentration of smallest particles is low, but their number concentration can be high causing health problems. However, when considering the air quality, the PM10 values are the most monitored and legislated. In Finland, the first EU directive was already implemented in August 2001. The daily 24 h average PM10 concentration was targeted to <50 μg m$^{-3}$, yet it



could be exceeded 35 times per year. This EU directive for PM10 values was entered into force in 2005. The most recent directive for PM2.5 with target value 25 µg m$^{-3}$ was given in 2010, and the value was entered into force in 2015. Besides the PM values, legislation considers also gaseous pollutants, some of which can interact with the formation of aerosol particles. In 2005, the given target value for $SO_2$ was 350 µg m$^{-3}$, and in 2010 it was decreased to 200 µg m$^{-3}$. $NO_2$, which is formed in combustion processes, is also regulated, and the target value was set to 40 µg m$^{-3}$ in 2010. The EU level directives can be

found in https://ec.europa.eu/environment/air/quality/standards.htm, (7.8.2020).

Laakso et al. (2003) reported the PM values from four measurement sites in Finland: Station for Measuring Ecosystem-Atmospheric Relations SMEAR I (Värriö), SMEAR II (Hyytiälä), SMEAR III (Helsinki), and Finnish Meteoritical Institute (FMI) at Pallas. They found that in 1999-2001 the average PM10 concentrations in rural area (SMEAR II) was 6.9 µg m$^{-3}$,

and in urban area (Helsinki, SMEAR III) 18.7 µg m$^{-3}$ which is at the same level as urban values in Leipzig in 2003 (Spindler et al., 2004). The reported urban and rural PM values were also in agreement with studies of PM concentration in Nordic locations such as Scandinavia (Forsberg et al., 2005). Long-term aerosol PM have been measured e.g. in Melpitz, Germany from 1993 to 2002 (Spindler et al., 2004) and the authors compared those values to the earlier ones from 1983 to 1992 from their measurement stations in same area (SLUG, Sachsisches Landesamt fur Umwelt und Geologie). Their measurement

location contained plenty of anthropogenic sources of pollutants and particles, e.g. industry and traffic, which were concluded to be the main source of PM10 and PM2.5 particles. They reported that the measured PM10 concentrations decreased from 1983-1990 level of ~70 µg m$^{-3}$ to ~50 µg m$^{-3}$ in 1994, to ~30 µg m$^{-3}$ in 1998, and to ~20 µg m$^{-3}$ in 2003. The decreases were explained by the German unification in 1990's, which was linked to the shutdown of the old coal factories and modernization of the power plants and heating systems of the houses in the area. In more recent study, Barmpadimos et

al. (2011) studied the influence of meteorology on the long-term (1991-2006) PM10 concentration trends in Switzerland in urban and rural areas. In 2002, their urban values (~25 µg m$^{-3}$) were quite close and rural values were somewhat higher (~20 µg m$^{-3}$) than the Nordic ones. They found a negative trend in PM10 in all environments on the given timescale of 14 years. The decreasing trend was ascribed to the new EU regulations for the PM and pollutant gas concentrations as well as the technology development in industry and vehicles (Barmpadimos et al., 2011). To give perspective for the concentrations, in

highly polluted areas in Beijing, China, the recorded PM10 values from 2004 to 2012, were ~138 ± 93 µg m$^{-3}$ for PM10, ~72 ±54  µg m$^{-3}$ for PM2.5 and 66 ± 56  µg m$^{-3}$ for PM1 (Liu et al., 2014).

Techniques for measuring aerosol mass concentrations have improved remarkably during the last decades (Van Dingenen et al., 2004). Most of the PM measurements have traditionally been done by offline gravimetric analyses where particle size

classes are separated e.g. by impactor (Laakso et al., 2003) or special high-volume samplers (Barmpadimos et al., 2011). The offline methods are quite laborious as their sampling time is up to few days and their weighing is manual. Thus, PM concentration is nowadays more commonly measured with instruments using on-line techniques, such as tapered element oscillating microbalance (TEOM) with the Continuous Ambient Particulate Monitor (Laakso et al., 2008) and Synchronized

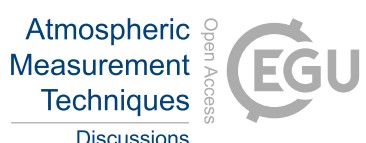

Hybrid Ambient Real-time Particulate monitor (SHARP) (Chen et al., 2018). Besides the direct mass measurements, the
particle mass can be calculated from on-line particle number size distribution measurements when particle shape and density
are approximated (Neusüß et al., 2000).

The aim of this work is threefold. First, we compare the gravimetric impactor, online mass analyzer SHARP and aerosol
mass concentration derived from number size distribution measurements to explore their applicability for continuous PM
measurements. Second, we report for the first time long-term (2005-2017) measurements and seasonal variations of PM10,
PM2.5 and PM1 concentrations in, SMEAR II, Finland, and explore the reasons for the overall concentration levels as well
as selected specific episodes. Third, we estimate the trends of the PM concentration during 2005-2017 for each season.
Quality controlled data on aerosol particle mass concentration in a boreal background station enable us to explore the role of
local, regional and global phenomena controlling the aerosol mass concentration in the area.

## 2. METHODS

### 2.1. Measurement station

The measurements were performed at SMEAR II located in Hyytiälä in southern Finland (61°51'N, 24°17'E; 181 m a.s.l.).
A photo of the homogeneous 58-year-old Scots pine stand surrounding SMEAR II is presented in Figure 1. Hyytiälä is a
rural background measurement site with low local anthropogenic emissions(Hari and Kulmala, 2005). The station is
equipped with instruments aiming for measuring continuously and comprehensively interactions between the forest
ecosystem and atmosphere (Kulmala et al., 2013). SMEAR II is part of the European Aerosols, Clouds, and Trace gases
Research Infra Structure (ACTRIS; http://www.actris.eu/, 7.8.2020). Most of the aerosol instruments, and all aerosol mass
measurement equipment, are located in "Hitu-hut" at SMEAR II (Figure 1c). There are total suspended particulates (TSP) or
PM10 design inlets for the different aerosol measurements on the roof of the hut.

### 2.2 Direct mass measurements with Cascade impactor

At SMEAR II, the collection of particulate samples with a gravimetric cascade impactor started in late 1990s. The impactor
has an unheated TSP inlet with stainless-steel tube (Table 1). The inlet is at 5 m height above the ground. The cascade
impactor has three stages with impactor cut-points at 10 µm (PM10), 2.5 µm (PM2.5) and 1 µm (PM1) (Dekati PM10
impactor)(Berner and Luerzer, 1980). The sample air flow rate during collection is 30 lpm. As collection substrates, 25 mm
polycarbonate membranes (Nuclepore 800 203) without holes are used. At the last stage a 47 mm teflon filter with 2 µm pore
size (R2P J047) from Pall Corporation is used. To prevent the bouncing back of the particles from the collection substrates,
the membranes are greased with Apiezon L vacuum grease diluted in toluene. The collected impactor samples are weighted
every two or three days to get the mass distribution. The samples are stored in freezer for occasional further analyses.



**2.3 Aerosol mass derived from the particle size distribution measurements**

The aerosol mass concentration for different size classes PM10, PM2.5 and PM1 can also be estimated by combining the number size distributions measured with Differential Mobility Particle Sizer (DMPS) and Aerosol Particle Sizer (APS) and calculating the mass by assuming that the particles are spherical and have constant density. This instrument set-up used at SMEAR II was developed in 2001 and is described in detail by Aalto et al. (2001). The DMPS inlet is placed on the roof of the hut at 8 m height and APS inlet at 5 m above ground level. It had ~10 min time resolution over the measured sizes.

Briefly, the twin-DMPS consists of a long and a short Vienna type Differential Mobility Analyzer (DMA) and two condensation particle counters (CPC, TSI 3025 and TSI 3775). At SMEAR II, the DMPS measures the number size distribution in the mobility diameter range of 3-1000 nm (Aalto et al., 2001). The APS (TSI 3320) measures the aerodynamic particle size distribution of particles with aerodynamic diameter within the range of 0.5-20 µm (Peters et al., 2006). To have comparable particle size distributions, we calculated the aerodynamic diameter from the mobility diameter

with the following equation:

$$\rho_0 d_a^2 = \rho_p d_m^2, \qquad\qquad [1]$$

where $d_m$ is mobility diameter, $\rho_p$ is the density of the particle, $d_a$ is the aerodynamic diameter and $\rho_0$ is the unit density of the particle (1 g/cm$^3$). The density of the particles is assumed to be 1.5 g/cm$^3$ (Saarikoski et al., 2005)). The mass of the particles measured with DMPS is calculated as:


$$m_{\mathrm{DMPS}} = \rho_p \cdot ((\pi d_m^3)/6) \qquad\qquad [2]$$

and with APS utilizing the mobility diameter:

$$m_{\mathrm{APS}} = \rho_p^2 / \rho_a \cdot ((\pi d_m^3)/6) \qquad\qquad [3]$$

The mass concentrations (PM1, PM2.5 and PM10) were then calculated by integrating over the corresponding size range. For PM1 only DMPS data was used.

$$\mathrm{PM}_i = \int_{dp0}^{1\mu m} N_{\mathrm{DMPS}} * m_{\mathrm{DMPS}} \, d d_m + \int_{1\mu m}^{i\mu m} N_{\mathrm{APS}} \ * m_{\mathrm{APS}} \, d d_m \qquad\qquad [4]$$

**2.3 On-line mass measurements with SHARP**

The Synchronized Hybrid Ambient Real-time Particulate Monitor SHARP (Thermo Scientific, Model 5030) is a real-time particulate monitor (Goohs et al., 2020). SHARP combines light scattering photometry and β–ray attenuation for continuous PM10 measurement. In SHARP the light scattering signal (nephelometer) is continuously calibrated against the beta attenuation mass sensor. The sample flow rate of SHARP is 16.7 lpm. The sampling temperature was fixed to 45 ˚C from

2012 to 2015 and to 35 ˚C after that. The sampling line is placed on the roof of the cottage at six meters above the ground





level. The measurements with SHARP started in 2012. SHARP measures on-line mass concentration with one second time resolution.

## 2.4 Correlations, bivariate fitting and long-term trends estimation by linear fitting

The correlations between the particulate mass derived from different instruments were calculated in Matlab by bivariate
fitting introduced by Cantrell et al., 2008. Before analysis, we removed clear outliers manually based on logbooks and mathematically in the Matlab from the data. The mass concentrations in SMEAR II were below 10 µg m$^{-3}$ in previous years (Laakso et al., 2003) and based on this information we filtered the mass values exceeding 20 µg m$^{-3}$. The long-term trends and statistical p-values were estimated by Matlab linear fitting tool.

## 3.   Results and discussion

### 3.1 Comparison between the mass measurement methods

Here, we present the correlations between the aerosol mass measurements with different instruments. The aim of the correlation analysis was to validate the mass measurement data. First, we studied the PM10 measurement results and coefficient of determination correlations ($R^2$) between the instruments. As seen from (Supplementary, Table S1), the correlation is high (~0.9-1) between the derived mass from DMPS+APS and SHARP measurements during 2012-2017. The
correlations are somewhat lower between the impactor and the two other methods. In summertime the correlations are worse, e.g. in 2016, which could be related to problems with the weighting of the impactor filters if there are some volatile compounds at summer. When excluding June and July in 2016, the $R^2$ values are mostly between 0.7-0.9. The impactor is, however, the only direct mass measurement at SMEAR II. Thus, in the next chapter, we test all the other methods against the impactor.
Here, we compare the measurement points averaged to impactor time resolution from all years for these three methods. The measurement points, (1:1)-line and a bivariate fit between the measurements performed by different methods are presented in Figure 2. Since, there is uncertainties with both axes data the bivariate fit was used. Figure 2 a presents the mass concentration measurements by SHARP vs DMPS+APS in 2012-2017 (SHARP measurements started in 2012). With PM10 data, the slope from bivariate fits between SHARP against DMPS+APS, SHARP against impactor, and DMPS+APS against
impactor are 0.8, 1.3 and 0.9 and their intercepts are 1.4, -1 and 0.7 µg$^{-3}$, respectively. With PM2.5 and PM1 the slopes of 0.91 and 0.8 and intercepts 0.8 and 0.01 µg$^{-3}$ between DMPS+APS and the impactor were obtained. Especially in the smallest particle size class, it seems that the on-line method gave somewhat smaller values compared to the impactor when the concentration was higher than a few micrograms (Figure 2). In the DMPS+APS method we assume constant density and spherical shape of the particles when we calculate their mass (see Methods) even though the particle composition (Heikkinen
et al., 2020), density and shape (Kannosto et al., 2008) are changing. This leads to higher uncertainty in the indirect



DMPS+APS mass calculations. The best agreement was found between the mass measurements of SHARP and impactor (Figure 2b) which were both designed to measure particulate mass.

## 3.2 The seasonal concentrations of PM10, PM2.5 and PM1

As the sources of PM vary between the different seasons, we analyzed the mass concentrations separately for each season.

Winter at SMEARII is defined to be from December to February (DJF), spring is from March to May (MAM), summer from June to August (JJA) and autumn from September to November (SON). The seasonal median concentrations of PM10, PM2.5 and PM1 from the impactor and DMPS+APS measurements at SMEAR II from 2005 to 2017 are presented in Figure 3a to 3f. In Figure 3g, the PM10 mass concentration results from SHARP are shown from 2012 to 2017. Figure 3h shows the median temperature measured at SMEAR II. To compare the results to earlier values at SMEAR II by Laakso et al. (2003),

we calculated also the mean values of the mass concentrations for the whole time period 2005-2017, and separately for 2005-2009 and 2010-2017 (Table 2, 3, and 4). In the springtime (MAM) the median mass concentrations were typically below the summer values or in the same range (Figure 3, green bars). The mean PM10 value was 5.4 µg m$^{-3}$ measured with the impactor and with the DMPS+APS for the years 2005-2017. The measured PM10 concentrations were well below the value (7.4 µg m$^{-3}$) reported by Laakso et al. (2003) during all years except 2006. In spring 2006, the forest fires in eastern Europe

had a clear effect on particulate mass at SMEAR II (Leino et al., 2014). In 2005-2009, the springtime PM10 concentrations were 5.8 µg m$^{-3}$ for the impactor and 6.3 µg m$^{-3}$ for the DMPS+APS method. After 2010, the values were 4.7 and 5.2 µg m$^{-3}$, respectively. More stringent EU air quality legislation might explain the springtime decrease in PM concentrations at SMEAR II.

Summer (JJA) median concentrations for all PM concentration are shown in Figure 3 with yellow bars. The highest

concentrations in years 2006, 2007, 2010, 2011, 2013, 2014 and 2015 were observed in the same years when the highest temperatures were recorded (Figure 3h). However, in summer 2006 (Leino et al., 2014) and 2010 (Heikkinen et al., 2020) there were also large-scale forest fires which could increase the PM concentration. In summer, the average PM10 concentration was 6.5 µg m$^{-3}$ (impactor) and 6.2 µg m$^{-3}$ (DMPS+APS) for years 2005-2017, 6 µg m$^{-3}$ (impactor) and 6.3 µg m$^{-3}$ (DMPS+APS) for years 2005-2009 and 6.9 µg m$^{-3}$ (impactor) and 6.2 µg m$^{-3}$ (DMPS+APS) for the years 2010-2017.


The higher concentration in summer was expected as the particle sources in summer season are mostly natural including particles formed from organic vapors emitted by the trees and plants (Heikkinen et al. 2020), as well as pollen (Manninen et al., 2009). Thus, EU regulations concerning industrial pollution and vehicle exhaust might not have a strong effect on (Ilyinskaya et al., 2017) summertime concentrations. In Table 3, the mean PM2.5 concentrations during 2005-2017, 2005-

2009, 2010-2017 and 1999-2002 are presented. Laakso et al. (2003) observed that during the summertime PM10 concentration was 7.2 µg m$^{-3}$ in SMEAR II in 1999-2000. The concentrations found in this study in 2005-2017 were slightly lower.


The median mass concentrations in autumn (SON) are shown in Figure 3 (pink bars) and the mean PM10 concentration was 4.5-5.1 µg m$^{-3}$ in 2005-2017 (Table 2). In comparison, Laakso et al. (2003) observed the PM10 concentration of 6.9 µg m$^{-3}$

for the years 1999-2002. Thus, the PM10 concentration decreased clearly from the value in 1999-2002 (Table 2). As seen in Tables 3 and 4, also the mean PM2.5 and PM1 concentrations decreased from 1999-2002 values. In autumn, the boreal forest starts to be less active in organic vapors production and anthropogenic emissions and long-range transport starts to have larger contribution to the PM concentrations. However, we did not observe remarkable decreasing of PM during 2010-2017. Instead, there was a higher concentration in 2014. The reason for the high PM mean concentrations in 2014 was probably a

six-month-long (from 31 August 2014 to 27 February 2015) eruption of the Bardarbunga volcano, Iceland, which Heikkinen et al. (2020) also noticed. During the eruption, high concentration of SO$_2$ and PM10 was measured over Europe, (Gislason et al., 2015; Ilyinskaya et al., 2017) which was predicted by the model which tracked the PM around Europe after volcano eruption by(Schmidt et al., 2011).

Winter (DJF) median concentrations for all PM concentration are shown in Figure 3 with yellow bars. There are three years

(2006, 2010 and 2011), which have higher median winter concentration than the others (Figure 3). These years 2006, 2010 and 2011 were colder (Figure 3h), which likely increased the particle emissions from heating procedures (likewise in Spindler et al. 2004, Barmpandimos et al., 2011). In wintertime, the residential combustion due to heating by wood has been observed to increase the PM concentrations e.g. in Switzerland (Barmpandimos et al., 2011). They also noticed that the concentrations in winter were similar during 1991-2002 even though the concentrations were descending in other seasons.

We suppose that most probably the anthropogenic sources (heating, traffic) of PM dominate in winter as the surrounding forest is in dormant state. Besides differences in the emissions sources, the dynamics (thickness) of the atmospheric boundary layer also influence to PMs. During wintertime, the boundary layer can be more than a kilometer shallower than in summer, and the anthropogenic pollutants become concentrated closer to the surface (Heikkinen et al., 2020b). Overall, the air quality at SMEAR II was very good throughout the years as the mean concentration for all size classes (2005-2017) were

from ~4 to 6 µg m$^{-3}$.

### 3.3 Seasonal long-term trends

Time slopes and intercepts for seasonal median mass concentration of each size class PM10, PM2.5 and PM1 are reported in Tables 3, 4 and 5. A time series of spring median concentrations of PM10 with their 25/75$^{th}$ quartiles are shown in Figure 4. A decreasing trend is observed in each measured size class. Linear regression fitted to the impactor measurement points give

us slopes -0.13(PM10), -0.14(PM2.5) and -0.13(PM1) µg m$^{-3}$ y$^{-1}$ and for DMPS+APS measurement -0.17(PM10), -0.17(PM2.5) and -0.13(PM1) µg m$^{-3}$ y$^{-1}$. The trends of spring-time data in all size classes except PM1 fit from DMPS results are statistically significant (p-values < 0.05). In spring, the trees start to produce organic vapors and there is gaseous SO$_2$ available mostly form the anthropogenic sources. Thus, note that springtime is the season with the highest particle number concentration due the new particle formation including ammonia, SO$_2$ and organics from forest has been observed (O'Dowd

et al., 2002). However, the mass of the recently formed particles is quite small compared to ambient PM1 (e.g. in winter).





After all, it is possible that the anthropogenic $SO_2$, PM10, PM2.5 emissions, have decreased due to new EU-legislation during these years as and thus we see the decreasing trend in all PM sizes in spring.

In summer, the forest is the most active and concentration of the organic species is the highest due to formation of secondary organic aerosol (Heikkinen et al., 2020). Additionally, pollen adds clearly to the mass concentration of PM10. The slope of

the fits was -0.04(PM10), -0.07(PM2.5) and -0.07(PM1) µg m$^{-3}$ y$^{-1}$ by impactor and -0.13(PM10), -0.10(PM2.5) and -0.08(PM1) µg m$^{-3}$ y$^{-1}$ by DMPS+APS. The latter were statistically significant with low p-values. The slopes for summer were a bit lower than the slopes fitted to the springtime data. Most of measured particles are natural e.g. pollen or originated from organic compounds emitted by the forest.

In the autumn we observed trends of 0.01(PM10), 0.01(PM2.5) and 0.01(PM1) µg m$^{-3}$ y$^{-1}$ by impactor and -0.08(PM10), -

0.09(PM2.5) and -0.06(PM1) µg m$^{-3}$ y$^{-1}$ by DMPS+APS.  If we consider the impactor method as the most reliable, it seems that the concentrations were steadier in autumn than other seasons. In the long run, the autumns have become warmer in Finland and the snow-covered period starts clearly later (Mikkonen et al. 2014). That could keep the level of organic compounds concentration higher longer in boreal forest which might be the reason that the particle concentration was not clearly decreasing in autumn. However, the result is not yet statistically significant, and we need deeper studies of this.


In the wintertime, the trend in particle mass is clearly negative. The slope is -0.08 µg m$^{-3}$ y$^{-1}$ by impactor for each size class and -0.16(PM10), -0.15(PM2.5) and -0.13(PM1) µg m$^{-3}$ y$^{-1}$ by DMPS+APS (Table 5). Here, the DMPS+APS resulted trends were the most evident (p-values below 0.05). In the winter, the organic precursors have minimum concentration and the collected particulate matter originates mostly anthropogenic sources e.g. traffic/industry/residential combustion nearby and

from long-distance transport of air pollutants. Actually, as introduced the boundary layer is clearly thinner at winter which concentrates the transported and nearby PM pollutants on the ground. It has been also studied that residential heating by wood combustion and is more significant in wintertime (e.g. Spindler et al., (2004) and (Jiang et al., 2019)) than in other seasons. It is quite interesting, that the DMPS+APS method gives more negative slopes than the impactor measurements. This could be a result of technical sampling efficiency or particle composition changing during the years. It is still an open

question. Nevertheless, the trends in particle mass for all size classes seem to be descending for spring, summer and winter at SMEAR II in 2005-2017.

### 4.    Conclusions

In this paper, the long-term, high-quality aerosol particulate mass PM concentration measurements at SMEAR II station in southern Finland were reported for the years 2005-2017. The direct mass concentration measurements and the size

distribution mass derived mass concentrations from aerosol size distributions measurements were compared in size classes PM10, PM2.5 and PM1. The mass concentration was measured by three methods: by the cascade Impactor (1) by DMPS and APS (2) and by SHARP (3). The data measured with the different instruments had mostly good correlation with coefficient



of determination ($R^2$) mostly over exceeding 0.8, with some exceptions. The lower correlation values were connected to measurement inaccuracies, especially related to the manual weighing of impactor filters. The different measurements were also compared to each other by the bivariate fitting method to detect any systematic differences. The methods showed excellent agreement with the slopes of 0.8-1.3 for PM10, 0.9 for PM2.5 and 0.8 for PM1. The data had also substantially low reasonable mass zero crossing points -0.99 – 1.4 µg m$^{-3}$ for PM10 and 0.8 µg m$^{-3}$ for PM2.5 and 0.01 µg m$^{-3}$ for PM1.

The measured PM values for all size classes were highest in summer (PM10 ~6.3 µg m$^{-3}$), second highest in spring (PM10 ~5.4 µg m$^{-3}$) and lowest in autumn (PM10 ~4.8 µg m$^{-3}$) and winter (PM10 ~4.8 µg m$^{-3}$). We identified anomalies in the seasonal mass concentrations were related to severe large-scale aerosol phenomena, such as big forest fires in spring 2006 in Eastern Europe, and volcanic eruptions taking place in Iceland in autumn/-winter 2014-2015. Otherwise, the annual summertime maximum concentrations can be connected to pollen, long-range transported aerosol, biomass burning, forest fires and higher temperatures (resulting in higher SOA concentrations) biogenic secondary organic aerosol produced within the boreal forest. However, source apportionment was not conducted within this work. Low wintertime temperatures resulted in higher mass concentration   most probably because of the concentrated PM concentration in thinner boundary layer from anthropogenic sources e.g. traffic/industry/residential combustion nearby and from long-distance transport.

We compared the mass concentrations with the earlier values from 1999-2000 reported by Laakso et al., 2003. The concentrations observed in our study were lower for the larger size classes (PM10 and PM 2.5) and similar for PM1. Importantly, we noticed, that the seasonal variation has changed throughout the years, as Laakso et al. (2003) observed highest PM10 and PM2.5 concentrations in springtime, while we observe them in the summertime.   The springtime had also the steepest decreasing trend in 2005 to 2017. This could indicate that the maybe the most important PM sources in boreal forest are shifting from anthropogenic emissions towards biogenic sources.

The results show the importance of long-term measurements for understanding atmospheric aerosol mass concentrations and factors controlling them. Comparison of particulate matter time series measured with different methods are valuable for data quality control purposes, as well as validating the applicability of the different methods. Therefore, we encourage to conduct extensive comparisons with different kind of particulate matter data series within each site and between different sites.  Only then we can see how natural and anthropogenic phenomena affect the PM concentration which is connected to adverse health effects via inhaling of these particles and their climate effect due to aerosol-cloud and aerosol-radiation interactions.

**Acknowledgements**

We want to thank the Academy of Finland (grant no. 272041, grant no. 311932) and the European Union's Horizon 2020 research and innovation program under grant agreements No.654109 and 739530 (ACTRIS). Also, we want gracefully thank the top-quality technical experts at SMEAR II: H. Laakso, M. Loponen, R. Pilkottu, T. Matilainen, P. Schiestl-Aalto and S. Rantanen.



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





Figure 1. (a) A photo of the surrounding region around SMEAR II. The area is dominated by boreal forest. (b) location

of SMEAR II (© OpenStreetMap contributors 2020. Distributed under a Creative Commons BY-SA License) and (c)

Hitu-hut for aerosol instrumentation.



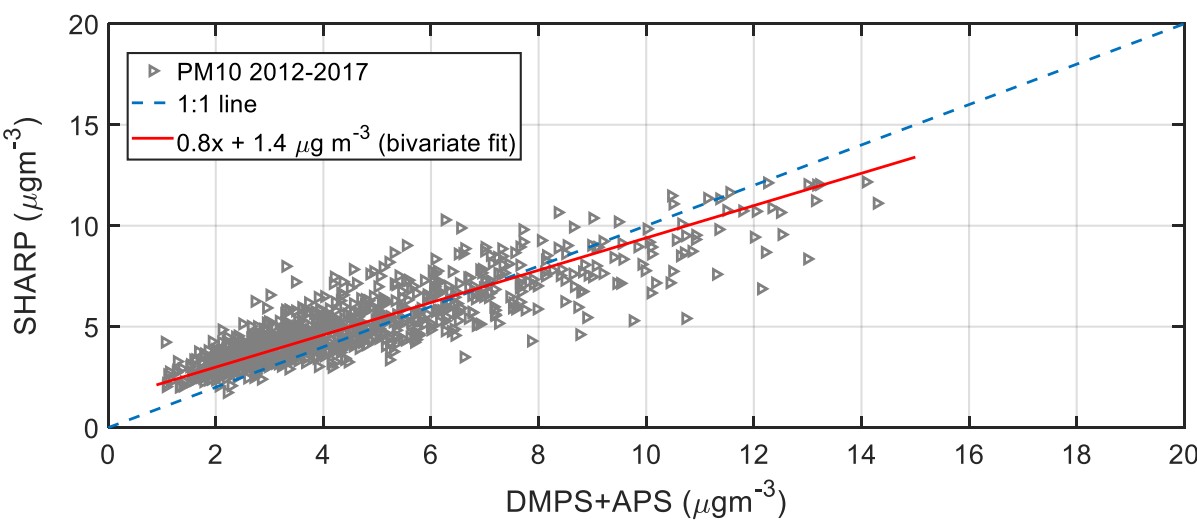

a)

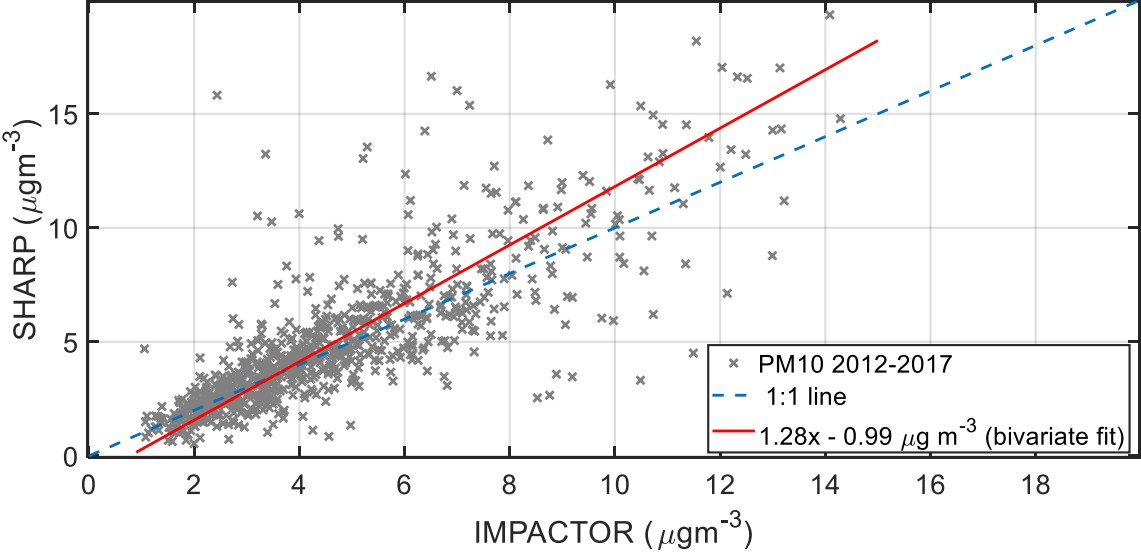

b)




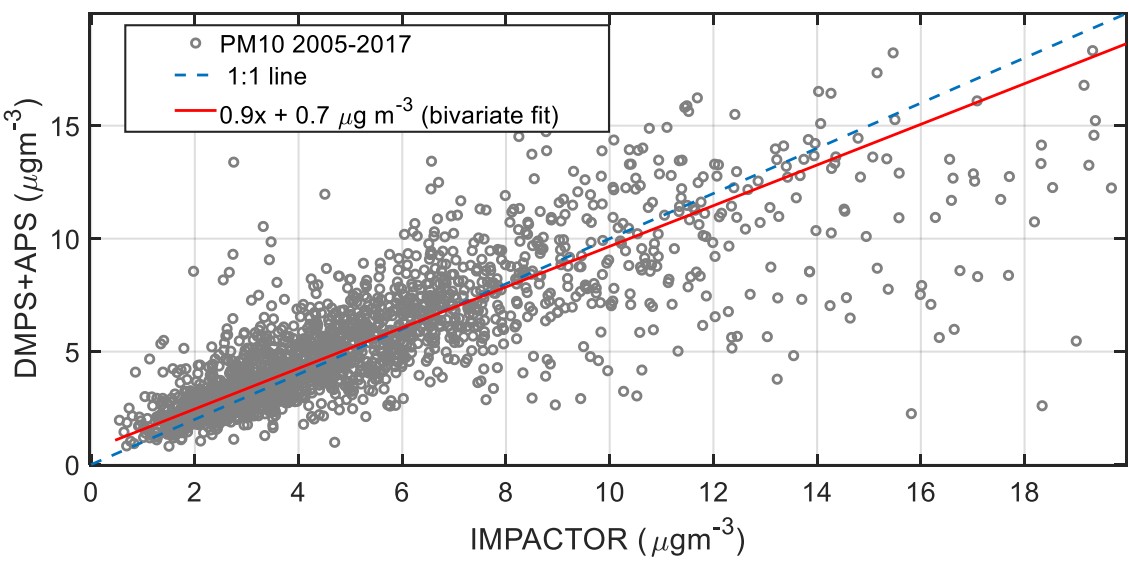

c)

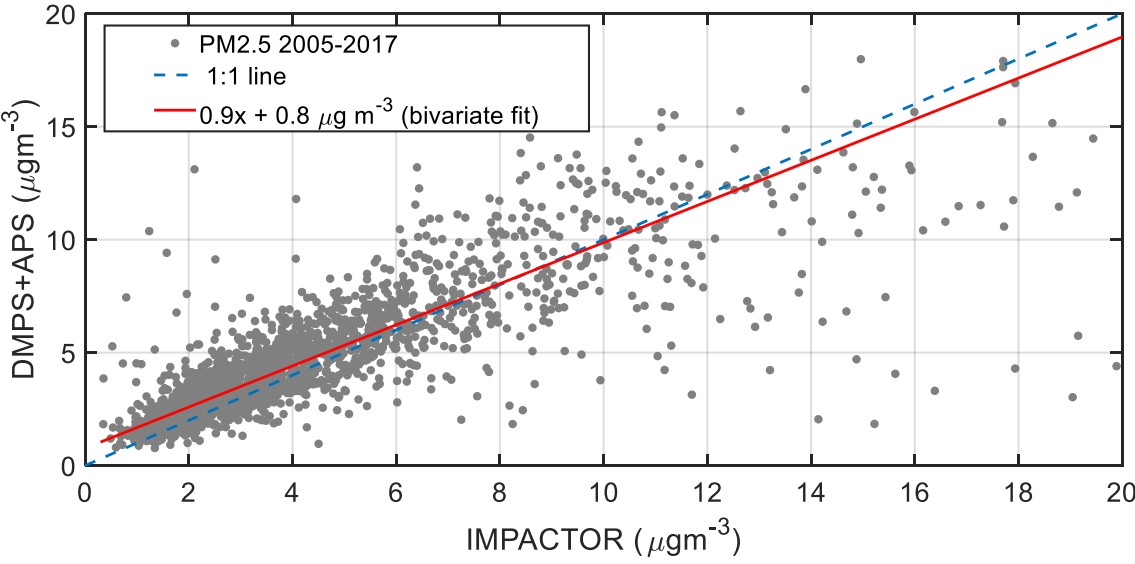

d)






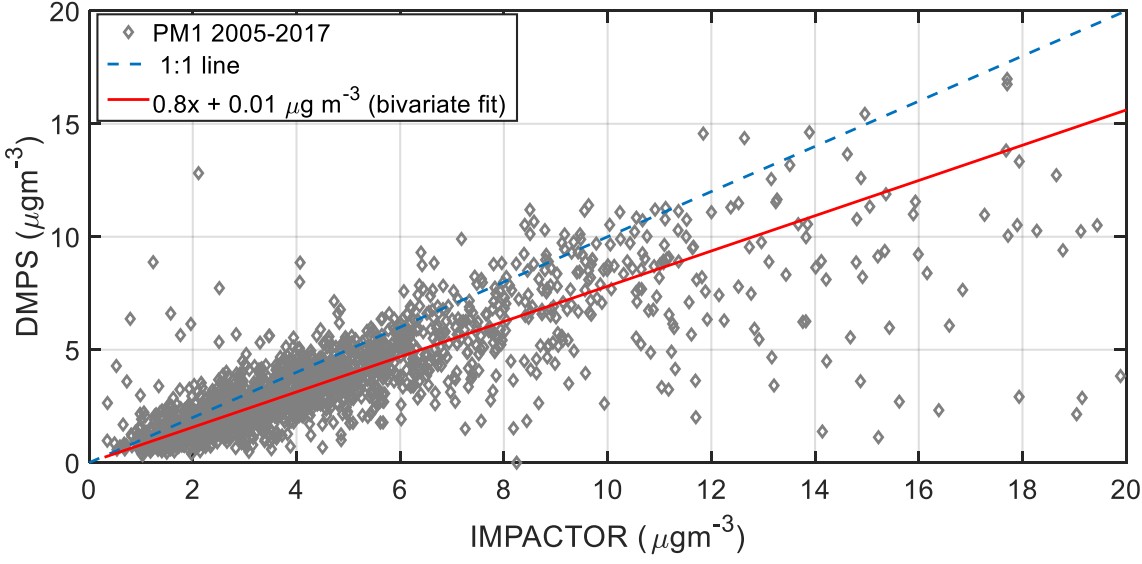

e)

Figure 2. Correlation between the different mass measuring methods a) PM10 from SHARP vs DMPS+APS b) PM10 from SHARP vs impactor and c) PM10 from DMPS+APS vs impactor d) PM2.5 from DMPS+APS vs impactor e) PM1 from
DMPS+APS vs impactor during their entire measurement period at SMEAR II until 2017. The red line represents the bivariate fit to the results with the slope and offset and the dotted one is the (1:1)-line. The points are averaged based on real impactor measurement points (2-3 days).





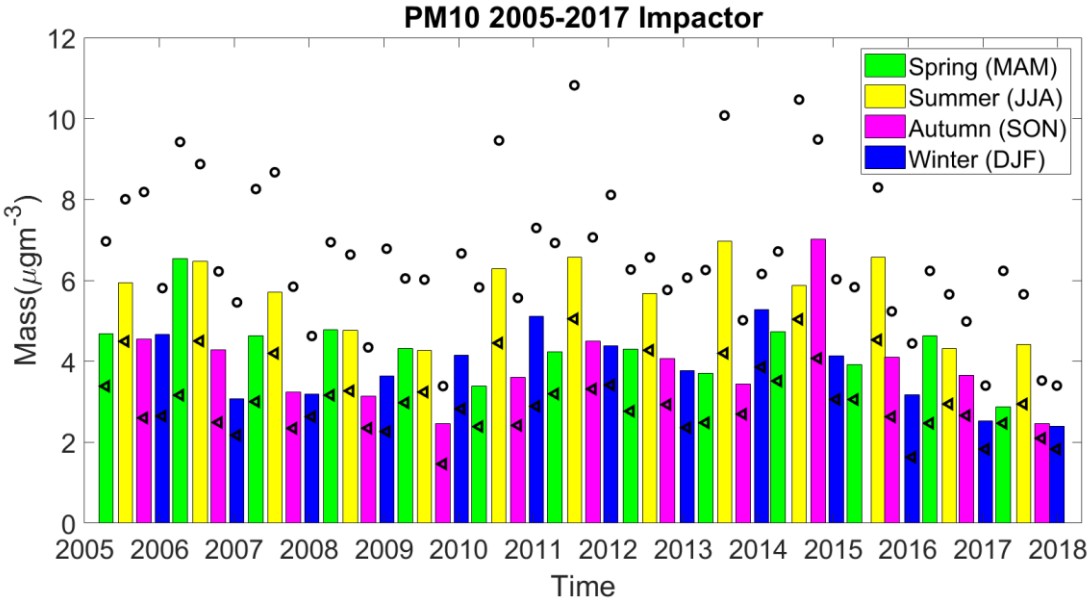

a)

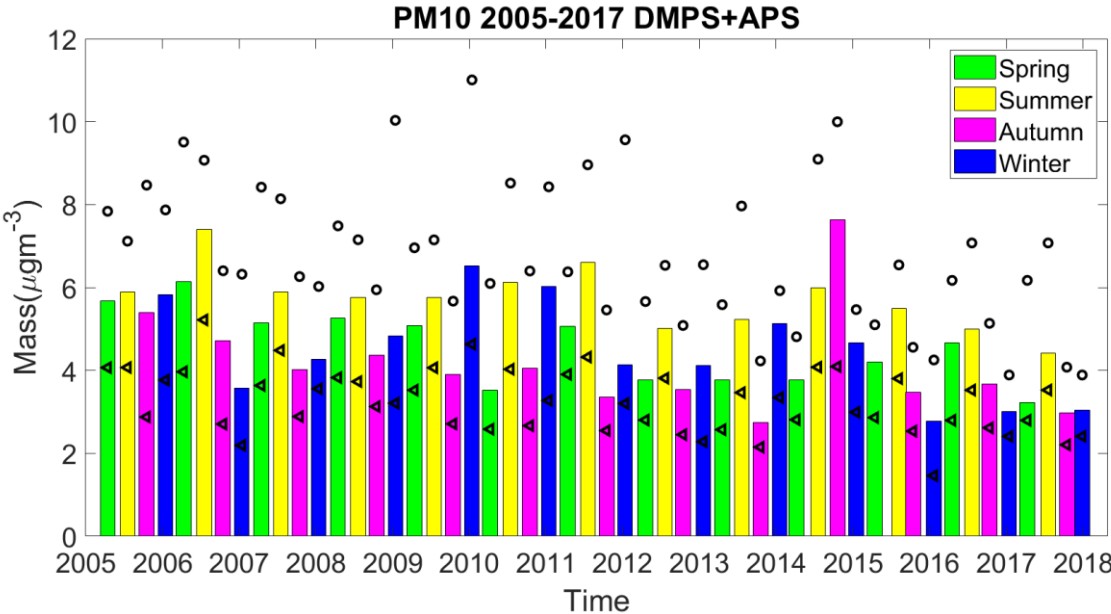

b)



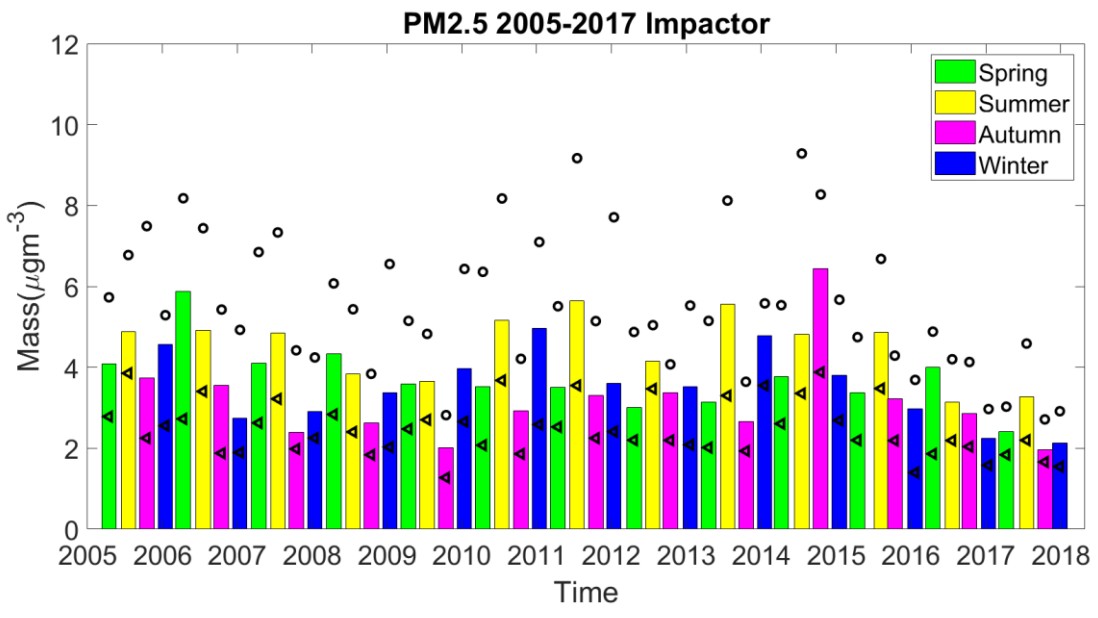

c)

d)



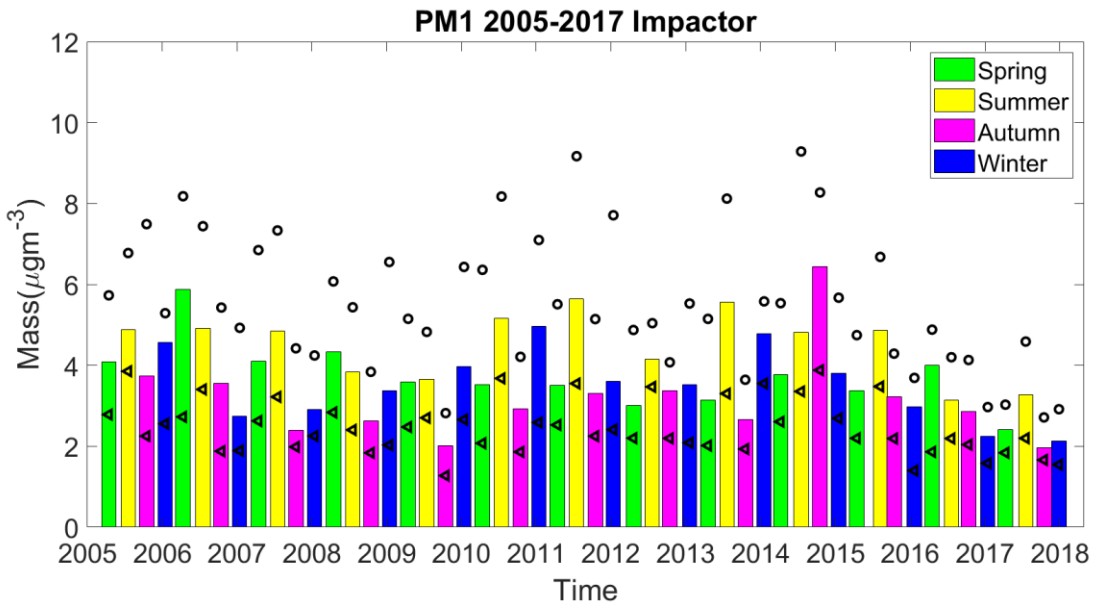

e)

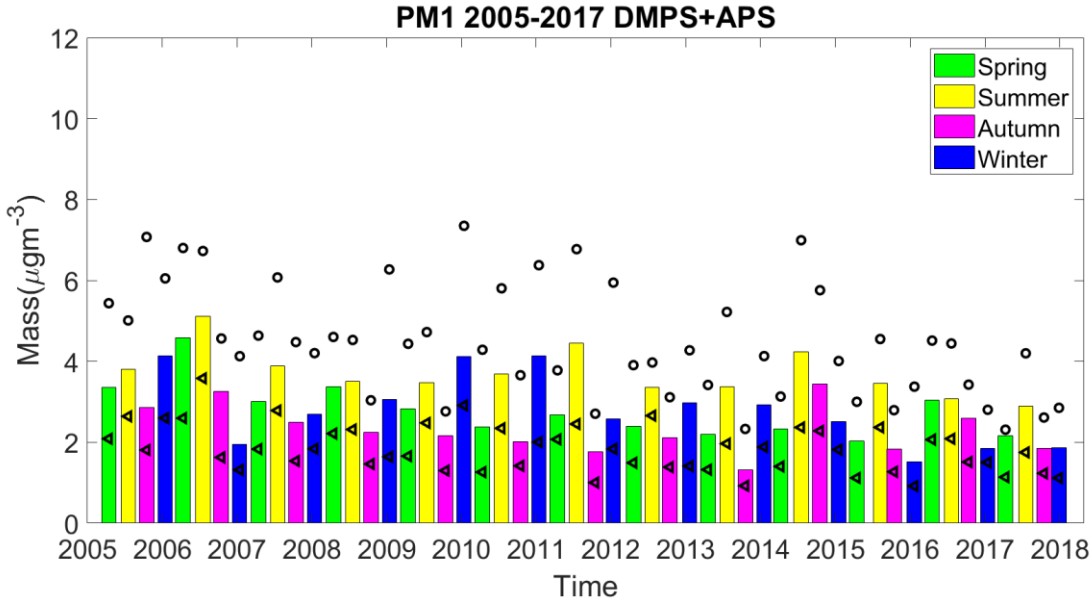

f)




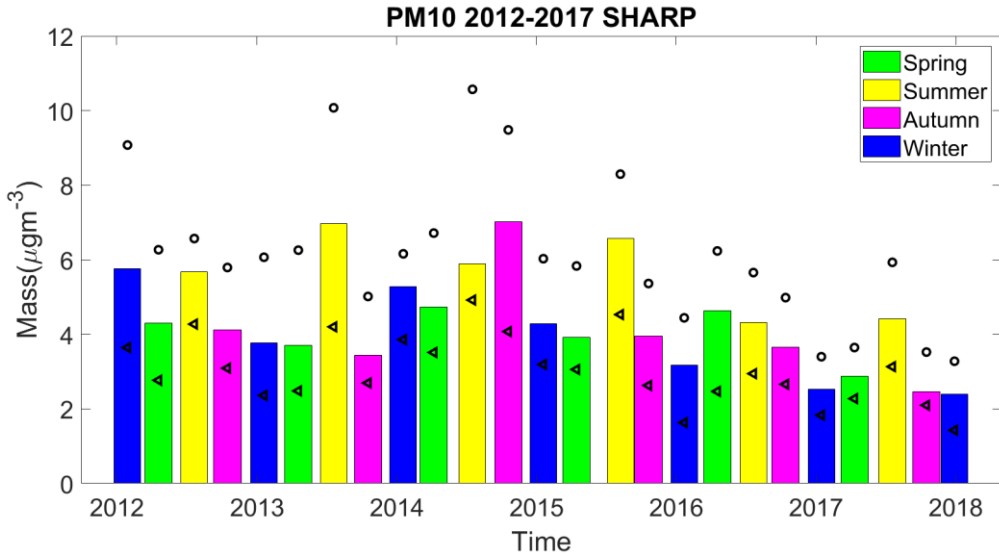

g)

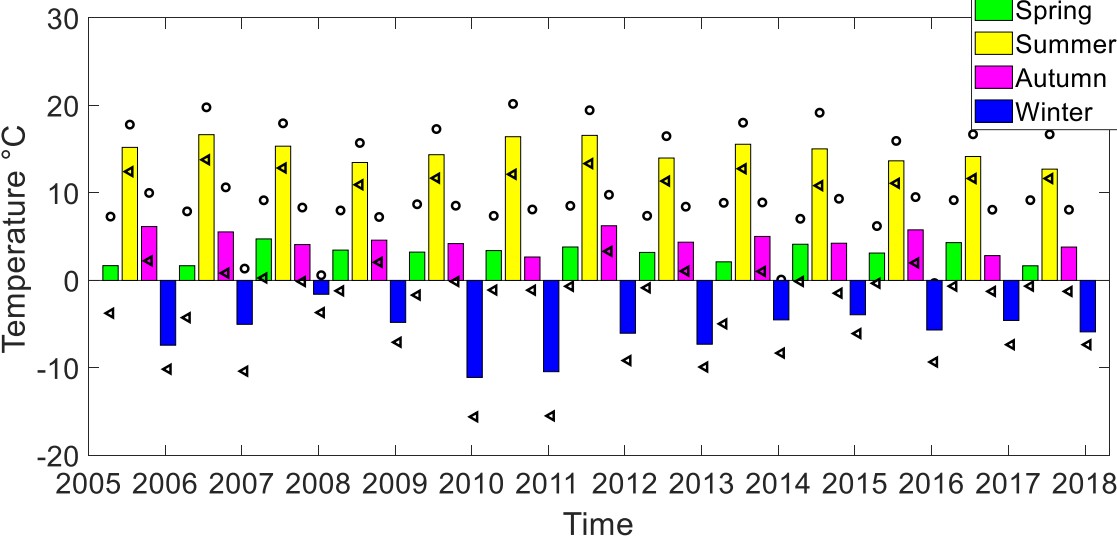

h)

Figure 3. Seasonal median PM10, PM2.5 and PM1 concentrations and their 25 and 75 quartile ranges measured with the impactor (a,c,e), DMPS+APS (b,d,f) and SHARP (g). Seasonal median temperatures at SMEARII and their 25/75 quartile ranges from 2005 to February 2018 (h).



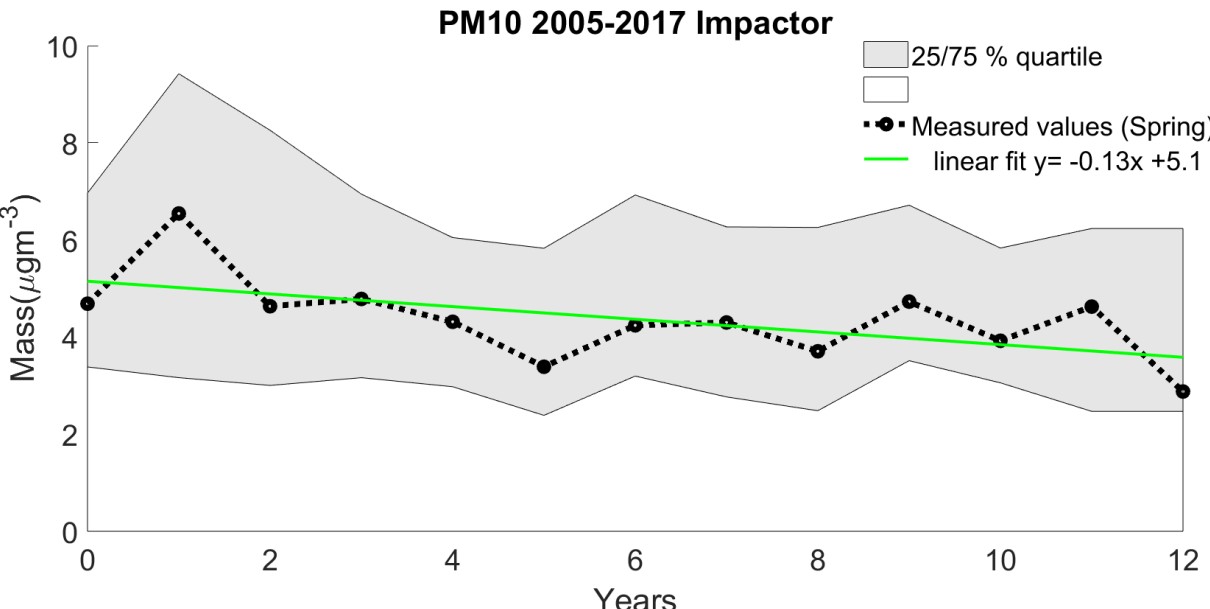


Figure 4. The springtime median PM10 concentrations and their 25/75 quartile ranges measured by the impactor from the year 2005 (0 in figure) to 2017. Linear regression line is fitted to the measurement points resulting the slope of -0.13 and y-axis interception of 5.1 µg m$^{-3}$.





Table 1. The measurement equipment used in this work, and detailed information of them  (UHEL=University of Helsinki)

| Method | Time period | Resolution | Manufacturer | Size range |
| --- | --- | --- | --- | --- |
| Impactor | 2005-2017 | 2-3 days | DEKATI Ltd. | PM1, PM2.5, PM10 |
| DMPS+APS | 2005-2017 | 30 min | UHEL and TSI Inc. | PM1, PM2.5, PM10* |
| SHARP | 2012-2017 | 1 min | Thermo Scientific | PM10 |

* calculated from size distribution measurements between 3nm – 20 µm.






Table 2. The average mean PM10 mass concentrations for each season at SMEAR II. The first number in each cell is estimation direct measurement from the impactor and second from the DMPS+APS measurements. *The average excluding the year 2014 (volcano eruption).

|  | PM10 (µg m$^{-3}$) 1999-2002 (Laakso et al., 2003) | PM10 (µg m$^{-3}$) 2005-2009 | PM10 (µg m$^{-3}$) 2005-2017 | PM10 (µg m$^{-3}$) 2010-2017 |
|---|---|---|---|---|
| Spring | 7.4 | 5.8 and 6.3 | 5.4 and 5.4 | 4.7 and 5.2 |
| Summer | 7.2 | 6 and 6.3 | 6.5 and 6.2 | 6.9 and 6.2 |
| Autumn | 6.9 | 5 and 4.5 | 5.1 and 4.5 | 7.6 and 7.4 |
| Winter | 5.1 | 4.4 and 5.6 | 5.1 and 4.5 | 4.5 and 4.5 |






Table 3. The average mean PM2.5 mass concentrations for each season at SMEAR II. The first number in each cell is estimation direct measurement from the impactor and second from the DMPS+APS measurements.

|  | PM2.5 ($\mu g\ m^{-3}$) 1999-2002 (Laakso et al., 2003) | PM2.5 ($\mu g\ m^{-3}$) 2005-2009 | PM2.5 ($\mu g\ m^{-3}$) 2005-2017 | PM2.5 ($\mu g\ m^{-3}$) 2010-2017 |
|---|---|---|---|---|
| Spring | 6.4 | 5.1 and 5.6 | 4.8 and 4.9 | 4.4 and 4.3 |
| Summer | 5.9 | 5 and 5.4 | 5.4 and 5.4 | 5.4 and 5.7 |
| Autumn | 5.7 | 3.8 and 4.6 | 4 and 4.2 | 4 and 3.9 |
| Winter | 5.1 | 4 and 5.3 | 4 and 4.8 | 4 and 4.3 |








Table 4. The average mean PM1 mass concentrations for four seasons for the certain years at SMEARII. The first number in each cell is estimation direct measurement from the impactor and second from the DMPS+APS measurements.

|  | PM1 (µg m⁻³) 1999-2002 (Laakso et al., 2003) | PM1 (µg m⁻³) 2005-2009 | PM1 (µg m⁻³) 2005-2017 | PM1 (µg m⁻³) 2010-2017 |
|---|---|---|---|---|
| Spring | 4.4 | 5.1 and 4.1 | 4.7 and 3.5 | 4.4 and 3.0 |
| Summer | 5.6 | 5.0 and 4.2 | 5.3 and 4.2 | 5.7 and 4.2 |
| Autumn | 3.6 | 3.8 and 3.4 | 4.0 and 3.0 | 4.1 and 2.7 |
| Winter | 3.8 | 4.1 and 3.8 | 4.1 and 3.4 | 4.0 and 3.0 |







Table 5. Linear regression fits and p-values for the time series of PM10, PM2.5 and PM1 mass concentrations for the four seasons at SMEARII during the years 2005-2017.

|  | SPRING | SUMMER | AUTUMN | WINTER |
|---|---|---|---|---|
| **PM10** |  |  |  |  |
| Impactor | -0.13x+5.1 (0.04) | -0.04x+6.0 (0.54) | 0.01x+3.8 (0.91) | -0.08x+4.3 (0.3) |
| DMPS+APS | -0.17x+5.6 (0.01) | -0.13x+6.5 (0.01) | -0.08x+4.6 (0.41) | -0.16x+5.4 (0.05) |
| **PM2.5** |  |  |  |  |
| Impactor | -0.14x+4.6 (0.01) | -0.07x+4.8 (0.27) | 0.01x+3.1 (0.89) | -0.08x+4.0 (0.23) |
| DMPS+APS | -0.17x+5.0 (0.01) | -0.10x+5.5 (0.04) | -0.09x+4.0 (0.21) | -0.15x+5.1 (0.07) |
| **PM1** |  |  |  |  |
| Impactor | -0.13x+4.6 (0.01) | -0.07x+4.9 (0.27) | 0.01x+3.1 (0.89) | -0.08+4.0 (0.23) |
| DMPS+APS | -0.13x+3.5 (0.46) | -0.08x+4.2 (0.04) | -0.06x+2.6 (0.24) | -0.13x+3.6 (0.04) |


