# Peer review of "Long-term aerosol mass concentrations in southern Finland: instrument validation, seasonal variation and trends"

_Atmospheric Measurement Techniques, 2020_

## Referee Comment (RC1) · Anonymous Referee #1 · 12 Mar 2021

Summary:

The manuscript presents a long ground-based dataset of aerosol mass concentrations at the SMEAR-II site in southern Finland. Mass concentrations are measured by integrating size distributions, by SHARP, and by using a standard gravimetric impactor technique. These measurements agree reasonably well for the 13-year record. Seasonal trends are discussed. While presenting long-term measurements is certainly valuable and worthy of publication, the text lacks second-order analyses that could make the conclusions less speculative and more defendable. Thus, I recommend publication only after major revisions.

Major Critiques

1. Size distribution measurements from DMPS+APS are one of 3 techniques used here to derive mass but these measurements are not explicitly shown in the manuscript. Combining these datasets is not trivial since each technique is fundamentally different and requires a knowledge of density and shape to be properly stitched. I think a necessary addition to this paper would be to show example distributions (at a minimum) or a statistical summary of the seasonal and annual variability in distribution. This is necessary for both method verification but also would add an additional layer of complexity to the paper that might help solidify some of the seasonal explanations presented later.

2. I think there is a lot of information content from the combination of these datasets that would be interesting to probe deeper and would significantly add value to the paper. For example, how does the DMPS+APS vs impactor regression vary seasonally, which may suggest that your constant density assumption breaks down in the presence of different dominant aerosol types? How has just the coarse-mode (i.e., PM10 minus PM1) varied annually and are those concentrations changing over time?

3. The Figure 2 regression analysis is a great first step but I'm not sure that a simple correlation coefficient and slope tell the full story. For example, panel B seems to show a concentration-dependent bias in the instrument responses. I suggest at least supplementing these regression plots by plotting instrument concentration ratios vs concentration. For example, for panel B plotting [SHARP / IMPACTOR] vs IMPACTOR would highlight the relative difference in the techniques for different mass loadings and illustrate the relative accuracy of SHARP at over a range of mass loadings.

4. There is a lot of speculation in the interpretation of trends sections (3.2 and 3.3) and discussion of each technique separately is very confusing. I recommend focusing the discussion on only the standard impactor technique here since the other techniques do not offer any additional information and correlated well. Also, one of the real values in this dataset is assessing the annual trends like in Figure 4. I suggest adding analogous

plots for the other seasons, and potentially for just PM1 (all seasons) so that these long-term trends can be highlighted.

Minor Edits:

Line Number

23 The statement "Temperature had a strong influence on the measured concentrations." is too speculative for the abstract and should be removed or the tone softened. I'm just not sure you can say this so strongly.

28 Remove "especially in winter", it is redundant.

46 You might want to include aircraft emissions (anthropogenic) and marine sea-spray aerosol (natural) in your list of aerosol types

134 What had a "∼10 min time resolution over the measured sizes"? Is the APS operated at 1Hz and averaged to 10 minutes, or are you referring to the DMPS?

139 This section was a bit confusing regarding whether you are working in aerodynamic or mobility diameter space. I think the first sentence should be flipped, "To have comparable particle size distributions, we calculated the aerodynamic diameter from the mobility diameter" should really be "To have comparable particle size distributions, we calculated the mobility diameter from aerodynamic diameter for APS measurements".

167 This 20 ug/m3 filtering seems arbitrary. Unless there is clear evidence that the measurement was contaminated by a local source or by an instrument operational issue, how do you justify simply removing the data? Can you provide statistics on how much data was removed by this filter, and the sensitivity of your results to this filtering approach?

174 For calculating PM10 from "DMPS+APS", how do you account for the overlap between the instruments in the 0.5-1.0um size range? Is there a stitching routine to produce a continuous size distribution? Are you choosing to use one instrument and

not the other? Is the APS data cut off above 10um so the data are directly comparable with SHARP and impactors?

178 replace "chapter" with "section"

226 The statement "In autumn, the boreal forest starts..." should be supported by references.

248 Why are you only showing SPRING median concentrations and fits in Figure 4? I suggest showing data and fits for each season.

252 Are there higher concentrations of SO2 in the spring compared to other seasons?

256 The sentence "After all, it is possible...." Sounds quite speculative. Can you provide references to support decreases in SO2, PM2.5 over this time frame related to EU policy?

259 This is speculative as well. Does your data show that there is a springtime peak in coarse-mode aerosols that could be ascribed to the presence of pollens?

296 "Otherwise, the annual summertime maximum..." is entirely speculative and should not be in the conclusion unless the tone is softened.

Table 1: Why not cut the APS distributions off at 10um so all the PM10 data are directly comparable?

Figure 2: Since the impactor measurements are your direct mass measurement that other techniques are being compared to, I would remove the "SHARP vs DMPS+APS" panel.

Figure 2: It looks like the range in SHARP values is very different between panels a-b. panel a varies from roughly 2-12ug/m3 while panel b varies from roughly 0-20ug/m3. Aren't these the same data?

Figure 3: The amount of data presented in Figure 3 is very impressive and certainly

worth publication. That said, since you have already established that there are good correlations between the different mass concentration methods, I suggest only showing impactor data, since it is the direct "standard" (panels a,c,e,h), in the main body of the paper and moving the rest to the SI.

Figure 3: I would consider adding a line representing the Laakso (2003) results for each of the PM1, PM2.5, and PM10 plots. The text continually emphasizes comparisons to this work so it would be helpful to graphically show those values in Figure 3.

---

## Referee Comment (RC2) · Anonymous Referee #2 · 8 Apr 2021

Overall comment: The manuscript shows a lot of work and many useful results for a certain community, but already by the title and the abstract it is doubtful that it is appropriate for a technical journal. The larger part of the manuscript is dealing with seasonal variation and trends of PM1, PM2.5, PM10 values in Southern Finland, often with respect to EU regulations, with various instruments and discussed in the style of a review paper. If the authors consider to keep it as a technical publication, it should be greatly revised and get much more focus on technical methods like bivariate fitting and its applicability for same or similar aerosol instruments, and other measurement sites. The extent of discussed examples of data can be reduced and shown, like in
some extent present in the text, with respect to limits and advantages of the validation, fitting, instruments uncertainties, maximum error and so on.

Having said that, I am not going into much detail in correcting sections, like seasonality and trends, which I suggest to be published as a separate publication or I would review in the next iteration. More specific comments:

Abstract, Line 26-28, ". . . can also be expected to be influenced. . ." too much "discussion style" in this sentence for an abstract section. Keep the abstract reduced to its purpose: what has been done and which results were obtained.

For my taste the introduction is too long and again mostly not refers to methodological or technical issues:

From line 36 to 91, quite long review of PM measurements.

Line 101 – 104: this could be actually the key part of the manuscript and get higher attention in the introduction.

P5, L120: All instruments should be described providing calibration/systematic and other uncertainties.

P5, L134: is 10 min time resolution the scanning cycle of the DMA?

P5, L143: the particle density is one of the most crucial parameters of this publication. I recommend to include more information about Saarikoski et al. 2005 already in the text. It should be mentioned that this reference is dealing only with one site in Finland. Preferably some geographical, boreal regions and the range of the aerosol densities should be given, at least for similar conditions.

P5, L146: the equation can be written without brackets

P5, L150: brackets can be removed

P5, L157: I suggest adding more information on the principles of the SHARP instrument. Does the nephelometer count single particles? Later on, it is said it is designed to measure mass directly. I failed to find Goohs et al 2009 on the web. Please include DOIs in all references!

P6, L163: Very brief section for the most technical part of a technical publication. Please provide more information on:

- the mathematical algorithms of removing outliers. E.g. if one point is above 20 and the neighboring ones are slightly below 20, is would not be an outlier in my opinion. The variance of data points is important.

- How many data points were filtered out this way?

- A summary of time constants of the three instruments in one place, and the common averaging time and the averaging method.

- If possible, name exact fitting algorithms, Cantrell et al gives only a review of them, as can be guessed from the title.

P6, L175: "somewhat lower" reads not very scientific. Do you have estimates for the possible volatile mass fraction? Please state correlation values including June and July. From the largest difference such effects can be estimated. Summarize not only correlation values but largest possible discrepancies between the three measuring methods/instruments.

P6, L182: "are" uncertainties. Bad formulation, rephrase this sentence.

P6, L185: I'm missing fitting uncertainty/variances of the slopes/offsets and data points, this also applies to the rest of the text and all figures.

P6, L187: somewhat smaller value? Which values? "R2 values". And remove "somewhat"

P6, L188, so, it seems that the value of 1.5 for density is used as a reference for this and some cited publications, to make it comparable in the case of Finnish boreal

sites. You should state it right at the introduction of the density, see L143. But does it make sense to report correlation values, if they are based only on the arbitrary fixed density? You could do the correlation based on the optimized value for density. Then the uncertainties of the Rˆ2 values are important. See P6, L185.

P6, L191: I find it contradictory "the best agreement was found between SHARP and impactor", but in the line 185 the slope for that is 1.3, which is higher than the others, also looking at the offsets it is not the lowest value. This whole section if difficult to follow, think about how the ordering of values, "optical to weight, weight to weight" methods can be named and sorted out better and separated from the interpretation. Maybe a table to summarize all that will help.

P10, L288 remove "over" in over exceeding.

P10, L290. Exactly, measurement inaccuracies should be described in the instruments section.